# A Comparative Study on the Effect of Ultrasound-Treated Apple Pomace and Coffee Silverskin Powders as Phosphate Replacers in Irish Breakfast Sausage Formulations

**DOI:** 10.3390/foods11182763

**Published:** 2022-09-08

**Authors:** Karthikeyan Palanisamy Thangavelu, Brijesh Tiwari, Joseph P. Kerry, Carlos Álvarez

**Affiliations:** 1Teagasc Food Research Centre, Ashtown, D15 DY05 Dublin, Ireland; 2School of Food and Nutritional Sciences, University College Cork, T12 E138 Cork, Ireland

**Keywords:** dietary fibers, technological properties, phosphate-reduction, clean-label meats

## Abstract

Ultrasound (US) technology can be used to improve the techno-functional properties of food ingredients, such as apple pomace (AP) and coffee silverskin (CSS), which can be used in meat products to enhance their quality. This study evaluated the changes produced by US-treated AP and CSS, when used as phosphate replacers, in the physicochemical properties of Irish breakfast sausages, i.e., their water holding capacity (WHC), cook loss, emulsion stability, proximate content, lipid oxidation, color, and textural parameters. Three sausage formulations with reduced phosphate concentrations were used to study the effect of US-treated AP and CSS, and an interactive relationship between US treatment and formulations using two-way ANOVA. The results showed that the addition of US-treated AP and CSS to all the formulations produced a significant interactive effect that increased the WHC (*p* < 0.05) and emulsion stability (*p* < 0.05), decreased cook loss (*p* < 0.05), and increased day 9 TBARS (*p* < 0.05) values of specific formulations. No significant changes were observed for the parameters of; color, texture, or proximate content values. Thus, this study demonstrated that the addition of US-treated AP and CSS improved the quality of phosphate-reduced sausages.

## 1. Introduction

Meat and meat products are widely consumed and play essential roles in consumer food choices, as they provide nutrients such as high-biological-value proteins, minerals (zinc, selenium, iron, and phosphorus), vitamins (B_12_ and other B complex vitamins), essential amino acids, and fatty acids [1]. However, a decline in processed meat consumption has occurred over the past decade for numerous reasons. One reason is the health concern that is linked to the composition of processed meat products, due to the presence of high salt contents, high fat levels, the presence of synthetic processing additives, etc. The current overuse of such processing ingredients have been linked to several profound health implications, including cardiovascular problems, obesity, kidney-related problems, and certain cancers [2,3]. In recent years, consumer demands for healthier, clean-labeled meat products containing no synthetic additives have forced processed-meat industries to reformulate their products to address consumer concerns.

Phosphates belong to one such synthetic additive grouping. They are added to processed meat products as emulsifiers, stabilizers, sequestrants, and thickeners and provide various functionalities (increased pH, decreased cook loss, improved water holding capacity (WHC), and enhanced textural and sensory properties) [4]. Alkaline phosphates work in synergy with sodium chloride (NaCl) in extracting myofibrillar proteins and improving meat products’ oxidative and microbiological stability [5], as they chelate metals and reduce water activity, respectively. The excess consumption of phosphates, i.e., higher than the acceptable daily intake (ADI) of 40 mg/kg for a healthy adult [6], is likely to cause fatal kidney-related problems in people with chronic kidney disorders [7]. Phosphate addition reduces calcium absorption, even in healthy individuals, leading to weakened bone strength [8]. These problems have led to the incorporation of alternative and natural agro-industrial co-products and/or techno-functional food ingredients into meat and meat products as a replacement for phosphate additives, thereby reducing the possible harmful effects on consumers and facilitating clean-labeling practices [4].

Dietary fibers are types of techno-functional ingredients that can be added to meat products with various health benefits, such as improving gastrointestinal function, reducing the incidence of breast and colon cancer, lowering cholesterol absorption, preventing cardiac diseases, and decreasing the risks of obesity [9,10]. Additionally, dietary fibers in meat products improve WHC, emulsion stability, cook loss, and textural and rheological characteristics [1,11]. Several studies reported these intrinsic techno-functional properties of several dietary fibers in meat products [12,13,14]. 

Apple pomace (AP) and coffee silverskin (CSS) are the two dietary fiber-rich agro-industrial co-products discussed in this study as phosphate replacers in Irish breakfast sausages. AP, obtained as a co-product from the apple juice industries, consists of pectin, polyphenols, vitamins, and organic acids. AP has a rich content of total dietary fiber (TDF) of 60% to 90% on a dry basis. AP is primarily insoluble, contributing to better rheological and WHC properties [15]. CSS is obtained as a co-product of the coffee roasting process. It is rich in proteins, minerals, and TDF on a dry basis (86%), and mainly insoluble. 

Both AP and CSS are reported to possess excellent antioxidant properties [16]. The rich source of dietary fibers in both AP and CSS makes them potential phosphate replacers in meat products. A previous study by our group [17], using an AP and CSS mixture in Irish breakfast sausages, showed that phosphates could be reduced to significantly lower quantities (up to ~80%) without significantly affecting the physicochemical and technological properties of these traditionally processed meat products. However, the complete replacement of phosphates negatively affected the quality of the sausages; hence, three optimized phosphate-reduced sausages were formulated. It was then hypothesized that applying a novel approach, modifying the ingredients to improve their techno-functional properties, could be used to further reduce or eliminate phosphates from sausage formulation. 

Ultrasound (US) technology is a non-thermal and green-processing technology that uses high-frequency sound waves to modify food ingredients physically and, thus, to modify their techno-functional properties, such as their water absorption capacity (WAC), oil absorption capacity (OAC), solubility, emulsification capacity, gelling capacity, swelling capacity, and foaming capacity [18,19]. The US works on the mechanism of cavitation, which is responsible for enhancing various food-processing techniques, including extraction, freezing, drying, emulsification, and inactivating pathogens [18]. From a previous study by our research group [20], when individual AP and CSS (10% *w*/*v*) suspensions were treated with US (20 kHz, 250 W) for 15 and 30 min, there was an improvement in the physicochemical properties, such as WAC, OAC, and viscosity values. This novel modification of ingredients can be used to address the aforementioned negative sausage qualities. 

It was noted that very few previous studies addressed the effects produced by the application of US-treated ingredients in food products. To our knowledge, the current study provides the first report on the impact of ultrasound-treated AP and CSS in phosphate-reduced meat products. Thus, the main aims of this study are to introduce the US-treated AP and CSS into the three previously optimized phosphate-reduced Irish breakfast sausage formulations and to assess whether the US-treated ingredients improved the technological properties of the phosphate-reduced sausage formulations, by comparing them with the properties of the sausage formulations with non-US treated ingredients.

## 2. Materials and Methods

### 2.1. Ingredient Preparation

Food co-products AP and CSS were responsibly sourced from Muns Agroindustrial S. LO. (Lleida, Spain) and Illy S. P. A. (Trieste, Italy), respectively. They were oven-dried at 40 °C to a constant weight before being ground into finely powdered ingredients using a laboratory mill (Perten Labmill 3100, PerkinElmer, Waltham, MA, USA). Samples were stored at 4 °C in sealable low-density polyethylene bags until further treatment. Both ingredients’ individual water suspensions (10% *w*/*v*) were prepared and stored at 4 °C (60 min) for complete sample hydration as a pre-treatment preparation. All solutions were treated using a high power (250 W, 20 kHz) US probe (UIP1000hdT, Hielscher Ultrasound technology, Teltow, Germany) for 30 min in a temperature-controlled setup (≤20 °C), consisting of jacketed glass beakers attached to a recirculating chiller (Lauda Brinkmann Ecoline RE104, Delran, NJ, USA). The selected US parameters used in this study were based on a previous study by Thangavelu, Tiwari, Kerry, and Álvarez [20], in which US treatment of AP and CSS for 30 min provided better improvement of functional properties, compared with US treatment for 15 min. All suspensions were freeze-dried (FD 80, Cuddon Freeze Dry, Blenheim, New Zealand), and the dried powders were stored separately at 4 °C in airtight plastic containers for use in further experimental analyses. Non-US treated ingredients were used with no other treatment than the initial drying step. 

### 2.2. Sausage Production

Sausages were prepared using pork lean meat (>90%) and fat obtained from three or four fresh pork loins (pH 5.3–6.0) that were purchased from a local butcher shop (Gleeson Butchers, Dublin, Ireland). The lean meat and back fat were minced using a meat mincer (Meat Grinder MG510, Kenwood, UK) and stored at 4 °C throughout all stages of production. The seasoning mix (without added phosphates) was purchased from Redbrook Ingredient Services Limited (Dublin, Ireland). The rusk and sodium tripolyphosphate (STPP) that was required for sausage manufacture was procured from AllinAll Ingredients (Dublin, Ireland). The sausages were prepared using the mixture composition (% of *w*/*w*) of pork lean meat (58.00%), pork back fat (20.35%), water/ice (13.45%), rusk (5.75%), seasoning mix (1.45%), and different combinations (1.00%) of STPP and/or phosphate replacers (US-treated AP and CSS) for each formulation. The formulations were as follows:
Formulation 1: 0.20% STPP + 0.22% AP + 0.58% CSSFormulation 2: 0.20% STPP + 0.00% AP + 0.80% CSSFormulation 3: 0.06% STPP + 0.94% AP + 0.00% CSS

These formulations were obtained from our previous validation process, which demonstrated the phosphate-replacing ability of AP and CSS in Irish breakfast sausages [17]. The formulations containing respective STPP and non-US-treated (natural) AP and CSS compositions were prepared using the above-mentioned sausage mixture composition and were considered as control samples for this study. (Because this study aimed to determine the impact of US on added recipe ingredients, traditional recipes were not included as a control, as our study previously demonstrated that such formulations performed better, in terms of functionality). The presence of US-treated AP and CSS in the treatment formulations distinguished them from the control formulations containing non-US treated AP and CSS. Sausages of ~10 cm in length and 23 mm in diameter per formulation, for both the control and treatment formulations, were prepared using a meat mincer fitter with a sausage filler (Meat Grinder MG510, Kenwood, UK). The prepared sausages were chosen randomly for further analysis. The functional properties of the sausage formulations containing US-treated AP and CSS were compared to the sausage formulations with non-treated AP and CSS. The prepared sausages (day 0) were retail-packaged using a padded black tray (h 197 mm × w 155 mm × d 30 mm; Silverstream Packaging Ltd., Cork, Ireland), heat wrapped using cling film (gas permeability −2.5 [g100μm]/[m2d]; 300 mm × 300 m, Prowarp, Bristol, UK) without the gas flushed. The sausage packs were then held under simulated retail chilled conditions (EXPO PT500, glass door upright display cooler, Mondial Framec srl, Mirabello Monferrato, Italy) at 3 °C to 5 °C for the duration of the storage trial and sampled at various specified time points. The sausages were analyzed for WHC (day 1 = the day after production day 0; 3 sausages per trial), cook loss (day 1; 3 sausages per trial), water mobility using low-field nuclear magnetic resonance (LF-NMR) (day 1; 3 sausages per trial), emulsion stability (day 1; 3 sausages per trial), texture profile analysis (TPA) (day 2; 3 sausages per trial), lipid oxidation (day 0, 3, 6, and 9; 3 sausages per day per trial), color (day 1; 3 sausages per trial), and proximate analysis changes, to examine the impact of US-treated AP and CSS in sausages. To achieve statistical validation, the entire study was performed in three independent trials on three different days and the obtained values were presented as the average of the outcome of the three trials. 

### 2.3. Proximate Analysis

Sausage batters per formulation were used to analyze the proximate composition of sausages. The protein percentage in the sausages was measured using the Dumas method, employing a LECO nitrogen content determiner (LECO FP628, LECO Corporation, St. Joseph, MI, USA) according to AOAC method 992.15 [21], using the nitrogen to a protein conversion factor of 6.25. Moisture and fat content were determined on the basis of AOAC 985.14 [22] and AOAC 2008.06 [23], respectively, using the SmartTrac5 rapid fat/moisture analyzer (SmartTrac 6, CEM Corporation, Matthews, NC, USA). Ash was determined on the basis of AOAC 920.153 [24], using a 550 °C Gellenkamp heating furnace (Gellenkamp, Cambridge, UK), and sodium chloride (NaCl) content was measured from the ash using the Bohr titration method on the basis of AOAC 935.47 [25]. According to AOAC 991.43 [26], the total dietary fiber content was measured using the ANKOM^TDF^ Dietary fiber analyzer (ANKOM Technology, Macedon, NY, USA). All experiments were carried out in three repetitions per independent trial, and the values were averaged.

### 2.4. WHC and Cook Loss

The WHC and cook loss of the sausages were measured using methods described by Lianji and Chen [27], with some modifications. Approximately 10 g of raw sausage (weight B) from three uncooked sausages per formulation per trial was taken and weighed in a 50 mL centrifuge tube followed by heating in a water bath at 90 °C for 10 min. The samples were then cooled to room temperature and weighed (weight C). The samples wrapped with cheesecloth were then placed in a centrifuge tube with 1/3 absorbent cotton wool and centrifuged for 10 min at 204× *g* (1000 rpm at 4 °C) in a Sorvall Lynx 6000 centrifuge (Fischer Scientific Ireland, Dublin, Ireland). The centrifuged samples were weighed and analyzed for WHC (1) and cook loss (2) values, using the following equations:(1)$$\mathrm{Water}\;\mathrm{Holding}\;\mathrm{Capacity}\;\left(\%\right)=1-\frac{\left(B-A\right)}{M}\times 100$$
where M is the total water content in sample meat calculated from the moisture values determined using the SmartTrac rapid fat/moisture analyzer (SmartTrac 6, CEM Corporation, Matthews, NC, USA).
(2)$$\mathrm{Cook}\;\mathrm{loss}\;\left(\%\right)=\frac{\mathrm{Initial}\;\mathrm{weight}\;\left(B\right)-\mathrm{Cooked}\;\mathrm{weight}\;\left(C\right)}{\mathrm{Initial}\;\mathrm{weight}\;\left(B\right)}\times 100$$

### 2.5. LF-NMR Analysis of Bound Water

The water mobility of raw sausage batters per formulation was measured using the LF-NMR as described by McDonnell et al. [28] using a Maran Ultra instrument (Oxford Instruments, Abington, Oxfordshire, UK) with a magnetic field of 0.5 Tesla and at a resonating frequency of 23.2 MHz. Ten g of raw sausage batter was taken in NMR tubes (15 mm diameter) and placed in a water bath (Model GD100, Grant Instruments Ltd., Cambridge, UK) at 25 °C for 1 h. Transverse measurements (T_2_) were obtained using a τ value of 150 µs and a relaxation delay of 5 s. Each measurement was obtained as the result of 16 scan repetitions. The T_2_ time distribution data (T_21_, T_22_, and T_2b_) and the population under the curve (P_21_, P_22_, and P_2b_) were obtained by applying multi-exponential fitting of the T_2_ relaxation data using the RI Win-DXP program (Oxford Instruments Molecular Biotools Ltd., Abington, Oxfordshire, UK).

### 2.6. Emulsion Stability

The emulsion stability of three sausages per formulation was measured on the basis of the method reported by Hughes, Mullen, and Troy [29]. The raw batter of approximately 25 g (exact weight recorded) was placed in a 50 mL centrifuge tube and centrifuged at 2958× *g* (1 min). Samples were heated in a water bath at 70 °C for 30 min and centrifuged at 2958× *g* (3 min). The supernatants were poured into pre-weighed crucibles and dried overnight at 100 °C. The pelleted samples were weighed. The volumes of total expressible fluid (TEF (%)) (4) and percentage fat exudate (5) were calculated as follows:(3)TEF=Weight of sample−Weight of pellet
(4)$$\mathrm{TEF}\;\left(\%\right)=\frac{\mathrm{TEF}}{\mathrm{Sample}\;\mathrm{weight}}\times 100$$
(5)$$\mathrm{Fat}\;\mathrm{Exudate}\;\left(\%\right)=\frac{\mathrm{Dried}\;\mathrm{Supernatant}}{\mathrm{TEF}}\times 100$$

### 2.7. Texture Profile Analysis (TPA)

Five raw sausages from each formulation were analyzed for hardness (N), chewiness (N), springiness (mm), and cohesion force ratio, based on Bourne [30]. The values of three sausage cores per replication per formulation were recorded. The sausages were cooked in a water bath for 20 min to 30 min at 73 ± 1 °C until the core was cooked at 70 °C on day 1 and cooled overnight. On day 2, the sausages were cored (14 mm diam. × 20 mm ht.) and force time deformation curves were obtained using an Instron universal testing machine, model 5543 (Instron (UK) Ltd., High Wycombe, UK), attached to a 500 N load cell and a compression anvil. The cores are axially compressed to 70% of their original height by the crosshead moving at the speed of 100 mm/min in a two-cycle compression test. The average values of three independent trials were produced. 

### 2.8. Color Evaluation

The color of three sausages per batch of each experimental formulation was measured using an UltraScan Pro (Hunterlab, Reston, VA, USA) dual beam xenon flash spectrometer, with a viewing port of 25.54 mm and illuminant D65, 10°. Calibration was carried out using a light trap (L = 0), and a standard white tile (L = 100; X = 88.69; Y = 93.58; Z = 100.45), covered in transparent PVC cling film to eliminate any color reading effect. To maintain reading uniformity, the sausages were packaged in PVC film for measuring color. The values were expressed as L* (lightness/darkness), a* (redness/greenness), and b* (yellowness/blueness) units. The total color difference (ΔE) between sausages was calculated using Formula (6) of Thangavelu, Tiwari, Kerry, McDonnell, and Álvarez [17].
(6)$${\Delta E}_{\mathrm{ab}}^{*}=\sqrt{{\left(L_{2}^{*}-L_{1}^{*}\right)}^{2}+{\left(a_{2}^{*}-a_{1}^{*}\right)}^{2}+{\left(b_{2}^{*}-b_{1}^{*}\right)}^{2}}$$

### 2.9. Lipid Oxidation

Lipid oxidation in the sausages was performed using a method based on Botsoglou et al. [31], with some modifications as demonstrated by Thangavelu, Tiwari, Kerry, McDonnell, and Álvarez [17]. Three sausage samples per formulation from days 0, 3, 6, and 9 of storage were analyzed via the TBARS method. Raw sausage batter (1.5 g) and 20 mL of milliQ water were homogenized using an Ultraturrax homogenizer (Labortechnik, Staufen, Germany) at 13,500 rpm for 30 s. Five mL of 25% cold trichloroacetic acid was added, followed by gentle stirring at 4 °C for 15 min, and centrifuged at 2498× *g* for 15 min (4 °C). The supernatant (3.5 mL) was mixed with 1.5 mL of 0.6% 2-thiobarbituric acid and heated in a water bath at 70 °C for 30 min. The tubes were cooled and measured at 532 nm using a UV-Vis spectrophotometer (Shimadzu UV-1700, Shimadzu Scientific Instruments, Columbia, MD, USA). The TBARS results were expressed as milligrams of malondialdehyde per kilogram of sausage (mg MDA/kg sausage).

### 2.10. Sausage Scoring System

The best overall sausage formulations, with improved physicochemical properties, were determined using the scoring method of Álvarez, Drummond, and Mullen [32]. Important parameters of interest that are mainly affected by the reduced phosphate concentration and that are affected by the inclusion of phosphate alternatives (US-treated AP and CSS) were considered as the inclusion criteria for the scoring system and each parameter was standardized using the following equations:(7)$$z=X-\frac{\mu }{\sigma }$$
(8)$$z=-\left(X-\frac{\mu }{\sigma }\right)$$
where z is the score value for a specific parameter, X is the mean value of the specific parameter, µ is the mean of all samples, and σ is the standard deviation.

Equation (7) was used for parameters where higher values were desired (Hardness, WHC, TDF), and Equation (8) was used for formulations where lower values were considered desirable (cook loss, TBARS day 9, and emulsion stability). The standardized values of all the parameters discussed were totaled to provide the overall grading score. In this study, the overall scores were divided by 2 to provide a better pictorial representation. This overall grading system was pictorially represented using a radar chart prepared with a Microsoft Excel sheet (Microsoft Corporation, Washington, DC, USA) to compare the scores of the sausage formulations.

### 2.11. Statistical Analysis

The entire study was repeated in three independent trials (control and treatments) over three days, and the replications were treated as blocks. The mean ± standard deviation results of the responses were obtained by averaging the triplicate values (five in the case of TPA) of all three independent trial values. The single factor impact and the interaction between the US-treated ingredients and different formulations were analyzed by two-way ANOVA, with US-treatment (non-US treated & US-treated AP & CSS), formulations (Formulations 1, 2 and 3) and US-treatment*formulations as the factors, using a Minitab^®^17.0 statistical software package, with the means of the data compared using Tukey’s comparison (*p* < 0.05). In addition, the means of the responses of each formulation (with US-treated AP and CSS) were compared with their respective formulation (with non-US treated AP and CSS), using a one-way ANOVA Tukey’s comparison (*p* < 0.05).

## 3. Results and Discussions

### 3.1. Proximate Composition Analysis

Proximate composition data for phosphate-reduced sausage formulations are presented in Table 1. The proximate content values for each formulation with US-treated AP and CSS, along with their respective formulations with non-US treated AP and CSS, were compared. As anticipated, the results of two-way ANOVA showed that the inclusion of US-treated AP and CSS in the sausage formulations did not produce a significant interactive impact on the proximate content of the sausage formulations. It was noted that the moisture, fat, protein, fiber, and salt content values of the sausages were nearly the same for all of the sausages formulations, irrespective of the AP- and CSS-type used. The values obtained were similar and supported by the validation work conducted in our previous research study [17]. The ash content of Formulations 1 and 2 (1.75% to 1.80%) with both non-US treated and US-treated ingredients was relatively high when compared with the ash content of Formulation 3 (1.64%). This was because of the reduced phosphate concentration and the absence of CSS in Formulation 3, as CSS is rich in inorganic minerals, such as phosphorus, potassium, calcium, magnesium, iron, and sulfur [33].

### 3.2. Emulsion Stability

The results of the emulsion stability analysis of the sausages were expressed using the TEF (%) and fat exudate (%) values, as shown in Table 1. In general, the lower the value of TEF (%) and fat exudate (%), the higher the degree of water and fat molecule binding within the meat matrix, indicating the higher stability of the emulsions [34]. The results of two-way ANOVA showed that US-treatment and the sausage formulations produced a significant interactive effect on TEF (%); however, no such effect was observed for fat exudate (%) values. This indicated that US-treated AP and CSS differently affected the TEF (%) values for different sausage formulations. When individual formulations were compared, statistical data of the one-way ANOVA analysis showed that the addition of US-treated AP and CSS to sausage Formulations 1 and 3 significantly reduced (*p* < 0.05) the values of TEF (%). This reduction in TEF (%) could be explained by the presence of US-treated AP, with significantly improved WAC and OAC produced by the US treatment for 30 min [20], which improved the water binding in the sausage batters. It was also noted that the TEF (%) values for Formulations 1 and 2 were lower for samples with non-US treated ingredients (Formulation 1- 8.2% ± 0.3%; Formulation 2- 7.5% ± 0.4%) and samples with US-treated ingredients (Formulation 1- 7.7% ± 0.1%; Formulation 2- 7.4% ± 0.2%) when compared with that of the Formulation 3 (non-US, 13.5% ± 0.6%; US, 10.6% ± 0.4%). This could be primarily due to the lower concentration of phosphate (0.06%) that was used and the absence of CSS in Formulation 3. The statistical one-way ANOVA analysis showed a significant decrease in fat exudate values (%) for Formulation 1 with US-treated AP and CSS, but was found to be insignificant for Formulations 2 and 3. This decreasing trend in TEF and fat exudate values was mainly due to the increase in the ingredients’ emulsifying capacity when treated with US. The emulsion stability of the formed emulsions depends upon the interfacial protein-membrane and oil-water droplet size [35,36]. Application of US led to the partial unfolding of the protein structure, thereby exposing the internal hydrophilic and hydrophobic protein groups that are present in the functional ingredients’ AP (~5% total protein content) and CSS (~18% total protein content) [20]. This, in turn, improved the adsorption of oil droplets to the protein structure, resulting in an improved emulsifying capacity of the ingredients [37] and resulting in improved emulsion stability of the phosphate-reduced sausage formulations.

### 3.3. Water Mobility Measured Using LF-NMR

LF-NMR relaxation data are generally used to measure water distribution and mobility in the meat muscle matrix, thereby helping to determine the various product qualities and attributes, such as WHC and drip loss, and sensory attributes [38,39]. The relaxation time distribution T_2_ data obtained from LF-NMR analysis can be differentiated into two or three curve compartments, each representing the different water types in the meat matrix [17]. The first compartment, T_2b_, with a time constant relaxation of 1 ms to 10 ms, represents inner protein-bound water. The second compartment, T_21_, with a 30 ms to 50 ms time constant, represents the myofibrillar active water. The third compartment, T_22_, with a time constant of 150 ms to 400 ms, represents the outer bound or extra-myofibrillar water that contributes to the drip loss. The population distribution under each curve (P_2b_, P_21_, and P_22_) demonstrates the amount of water present in the specific relaxation time distribution curve [17].

The results of the relaxation time (T_2b_, T_21_, and T_22_) and the population distribution (P_2b_, P_21_, and P_22_) for the sausage samples are presented in Table 1. The results of two-way ANOVA showed that US-treated ingredients and different sausage formulations had a significant interactive effect on population distribution values (P_2b_, P_21_, and P_22_), which was not the case for relaxation time (T_2b_, T_21_, and T_22_). It was observed from the one-way ANOVA results that the population distribution of relaxation curve T_2b_ (P_2b_) of Formulations 1 and 2 had a significant decreasing effect when US-treated AP and CSS were added. It was previously reported that the water distributed under the T_2b_ curve was generally unaffected by any mechanical stress [28], but in our study, changes (*p* < 0.05) were observed due to the addition of AP and CSS, confirming the results of our previous study [17]. This was supported by the significant decrease in the values of T_2b_ for all three formulations when US-treated AP and CSS were added. The values of P_21_ for Formulations 1 and 2 showed that the addition of US-treated AP and CSS increased population distribution (*p* < 0.05), compared with their respective formulations with non-US-treated AP and CSS, thereby confirming the increased WHC results that were observed. However, a decreasing trend in P_21_ values (*p* > 0.05) was observed in the case of Formulation 3. The results of P_22_ showed that the addition of US-treated AP and CSS reduced the values in Formulations 1 and 2 that were not sufficiently significant to support the impact of US application. As with the results of P_21_, the P_22_ values of Formulation 3 followed an opposite trend compared with that of Formulations 1 and 2. This increase in P_21_ and the decrease in P_22_ values for Formulations 1 and 2 demonstrated that water from the outbound water matrix had moved to the entrapped water matrix. This demonstrated that the water mobility in the meat matrix was still mainly influenced by the phosphate concentration. The addition of US-treated AP and CSS to phosphate-reduced sausage formulations positively influenced the water mobility of Formulations 1 and 2.

### 3.4. WHC and Cook Loss

WHC can be defined as the ability of meat to maintain its inherent water content when subjected to mincing, cutting, pressing, and during storage and transport, thereby determining its acceptability, weight loss, cook loss, and sensory traits for consumption [40,41]. The mean values of the WHC for all of the experimental sausage formulations are presented in Figure 1a. In comparing the formulations, it was observed that the WHC value of Formulations 3 with non-US treated AP and CSS (76.9% ± 1.2%) was the lowest, when compared to Formulations 1 (82.8% ± 0.1%) and 2 (82.3% ± 0.7%) with non-US treated AP and CSS. Similarly, the lowest WHC value was observed for Formulation 3 with US-treated AP and CSS (81.1% ± 0.6%) when compared with other formulations with US-treated AP and CSS (Formulation 1- 85.0% ± 0.3%; Formulation 2- 85.3% ± 0.6%). This observation was due to the reduced phosphate concentration in Formulation 3, as phosphates significantly improve the WHC in sausages [42]. The results of two-way ANOVA showed that the addition of US-treated AP and CSS to sausage formulations had a significant interactive effect (*p* < 0.05) on the WHC in sausages. It was observed that the WHC values of Formulations 2 and 3 with US-treated ingredients significantly increased, when compared with their respective formulations with non-US-treated ingredients, which was not the case with Formulation 1, where an insignificant increasing trend was observed. This increase in WHC was due to the physical modification induced in the ingredients by the cavitation mechanism produced by the US application [18]. This alteration in the ingredient matrix opened the structure of the treated ingredients, thereby increasing their electric charge and exposing their internally hidden hydrophilic groups to water molecules, resulting in an increased water absorption capacity (WAC) of US-treated AP and CSS, as demonstrated in our previous study [20], thereby increasing the WHC of the sausage formulations.

Results of two-way ANOVA showed that cook loss values were significantly influenced by the interactive effect of the sausage formulations and US-treated ingredients. Figure 1b shows that the introduction of US-treated AP and CSS reduced the cook loss values of the sausage formulations. Statistical analysis showed that this effect was significant (*p* < 0.05) for Formulation 2 (Con, 9.1% ± 0.4%; US, 7.6% ± 0.3%), Formulation 3 (Con, 13.1% ± 0.8%; US, 9.6% ± 0.3%), whereas it was insignificant for Formulation 1 (Con, 9.1% ± 0.1%; US, 8.3% ± 0.4%). The reduction in cook loss values can be explained because cook loss is inversely related to WHC. The observed increase in WHC produced by the tight binding of water molecules resulted in decreased free water molecules in the meat matrix, thereby reducing the cook loss values. It was also observed that the cook loss values for Formulation 3, irrespective of whether US-treated or non-US treated ingredients were employed, were higher when compared to those for Formulations 1 and 2. Again, this is due to the lower STPP concentration (0.06%) present and the inability of the AP to match the STPP level that was utilized. These analyses showed that the introduction of US-treated AP and CSS improved the quality of phosphate-reduced sausage formulations and positively impacted the WHC and cook loss values.

### 3.5. Color

The instrumental color analysis results of raw sausage formulations are presented in Table 2. Based on our group’s previous study, significant changes in the color parameters of AP and CSS powders were observed when they were treated with US for 30 min [20]. However, when integrated into a sausage batter, according to the one-way ANOVA results, US-treated AP and CSS produced no significant changes in the mean L* and b* values of the sausage formulations, compared with their respective formulations with non-US treated AP and CSS. The introduction of US-treated AP and CSS produced a significant increase (*p* < 0.05) in a* values only for Formulation 2. This increase in sausage redness was due to the color change produced in the AP and CSS, due to US treatment and freeze-drying. In addition, the results of two-way ANOVA showed that US-treated ingredients and formulations did not significantly interact with the color values.

The readings of ΔE, measuring the visual color difference between the formulations with non-US treated ingredients and their respective sausage formulations containing US-treated AP and CSS, were as follows: Formulation 1, 0.82, Formulation 2, 0.60, and Formulation 3, 0.89. According to Mokrzycki and Tatol [43], no color differences will be observed between samples by inexperienced observers if the samples possess ΔE values between 0 < ΔE < 2, and a visible color difference will be observed by inexperienced observers if values of over ΔE > 2 are achieved. Thus, no visible color differences were observed between the sausage formulations with US-treated AP and CSS and their respective formulations with non-US treated AP and CSS.

### 3.6. Texture Profile Analysis (TPA)

The results of texture parameters for sausage formulations consisting of hardness, springiness, chewiness, and cohesive force values are presented in Table 2. The results of two-way ANOVA analysis of texture parameters showed no significant interactive differences between the sausage formulations and US-treated AP and CSS. A previous study by Thangavelu, Tiwari, Kerry, McDonnell, and Álvarez (2022) [17] showed that an increase in AP and CSS concentration reduced the hardness and chewiness values in sausage formulations containing higher STPP. Thus, the introduction of US-treated AP and CSS to improve the textural characteristics of phosphate-reduced sausage formulations did not produce any significant changes. Although there was a significant increase in the WHC and emulsion stability in sausages formulated with US-treated AP and CSS, the data generated did not influence sausage textural properties. This finding demonstrates that sausage texture is primarily dependent on STPP concentration, despite US-treated AP and CSS employment in the sausage formulations.

### 3.7. TBARS Analysis of Sausages

Lipid oxidation measurements in the sausage formulations were expressed in milligrams of malondialdehyde produced per kilogram of sausage (Table 2), using the TBARS method. The results of two-way ANOVA, measuring the interaction effect between the US treatment and the formulations, did not produce any significant interaction effect on the TBARS values on days 0, 3, and 6. However, the effect was significant (*p* < 0.05) for the TBARS value on day 9, indicating that the US-treated ingredients produced a different effect on different formulations. The results of one-way ANOVA showed that the TBARS values (measured on storage days 0, 3, and 6) of the sausage formulations containing US-treated ingredients were not significantly different from their respective formulations with non-US treated ingredients. However, in the case of day 9 TBARS values, it was evident that sausages containing US-treated AP and CSS had higher (*p* < 0.05) values for Formulations 2 and 3 (*p* < 0.05), compared with their respective formulations with non-US treated ingredients, with an exception for Formulation 1. However, the increase in TBARS value ranges was well within the detectable threshold limit of 2.0 mg MDA/kg to 2.5 mg MDA/kg [44], the highest upper degree where no off-flavors are expected to appear, and the formation of dangerous free radicals was in control. Comparing the formulations, it was observed that Formulations 1 and 2 with US-treated AP and CSS had lower day 9 TBARS values, compared with Formulation 3 with US-treated AP and CSS, which was expected because of the lower STPP concentration (0.06%) of Formulation 3. This may be explained by the fact that phosphates are known to be excellent antioxidants [4]. However, this was not observed for the TBARS values measured on days 0, 3, and 6. This control in the TBARS values was due to the addition of AP and CSS, which are excellent antioxidants [16], even after the US treatment. Therefore, the addition of US-treated AP and CSS to improve the quality of phosphate-reduced sausage formulations resulted in an increased value of lipid oxidation on day 9.

### 3.8. Sausage Formulation Scoring

The results of the sausage scoring system employing important physicochemical product properties are shown in Figure 2. The scores of phosphate-reduced sausage formulations improved with the US-treated AP and CSS, compared with the addition of non-US treated AP and CSS. The total scores (divided by 2) of the sausage formulations with non-US treated and US-treated AP and CSS, respectively, were as follows: Formulation 1: 0.75 and 1.29; Formulation 2: 1.00 and 2.93; and Formulation 3: −4.05 and −1.94. These scores indicated the quality improvement in the sausage formulations by the addition of US-treated AP and CSS. It was noted that hardness and TBARS day 9 values had more weight on these scores, along with the WHC, cook loss, and emulsion stability. The TDF can also be validated as a more important trait with reduced-phosphate content. Although both AP and CSS contain almost the same TDF %, the concentration of STPP added and their interaction with STPP produced differences in formulation scores. On reflection, the scores of Formulation 2 containing 0.80% CSS and 0.20% STPP had the highest TDF values, whereas Formulation 3 containing 0.94% AP and 0.06% STPP recorded the lowest TDF values. In summary, STPP concentration still had an influential impact on the TDF scores; therefore, Formulation 2, which possessed the higher TDA score, can be regarded as the best sausage formulation. The TDA scores also showed that CSS had a better phosphate-reducing capability when compared with AP. 

## 4. Conclusions

Interpretation of the significant sausage properties influenced by phosphates, such as WHC, cook loss, and emulsion stability, showed that US-treated AP and CSS positively affected phosphate-reduced sausages, compared with those generated using non-US treated AP and CSS, thereby improving their quality. In addition, changes in these properties were significantly influenced by the interactive relationship effect produced by the addition of US-treated ingredients and different sausage formulations, implying that different formulations reacted differently to the addition of US-treated AP and CSS. The findings from this study clearly showed that STPP could be reduced from 0.5% to 0.2% with improved product quality attributes when US-treated AP and CSS were added to Irish breakfast sausage formulations. Further formulation grading analysis showed that Formulation 2, containing only CSS as an STPP replacer, had a better score for physicochemical properties than Formulation 3, which contained only AP. Thus, US-treated CSS is the best STPP-replacer, compared with AP, leading to the conclusion that Formulation 2 is the best-optimized formulation. However, increased TBARS values were the study’s primary concern, although the values observed were within acceptable thresholds, as determined scientifically. This could be further addressed by following various approaches in future studies, by increasing the percentage of US-treated AP and CSS (up to ~2%) in sausage formulations or by increasing the frequency of US treatment over 20 kHz, or changing the mode of US treatment (ultrasonic bath), with more emphasis given to the sensory and shelf-life characterizations in either of the discussed possibilities.

## Figures and Tables

**Figure 1 foods-11-02763-f001:**
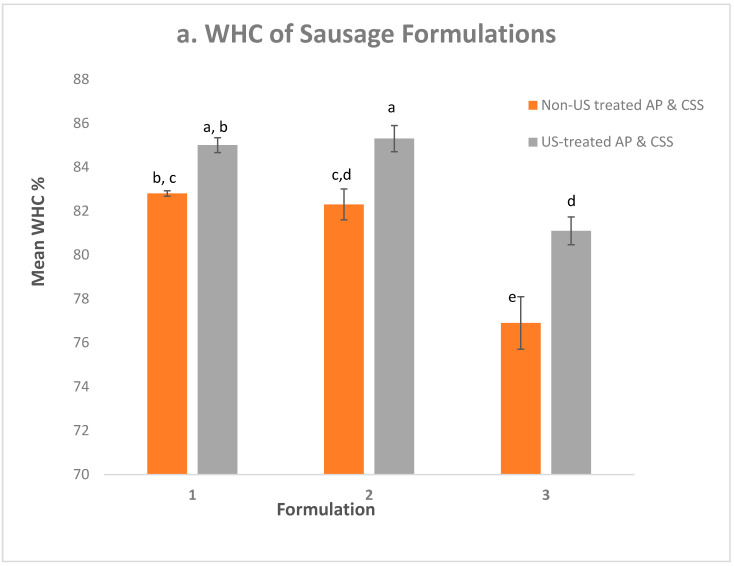
Bar chart comparing (**a**) WHC and (**b**) cook loss values of sausage Formulations 1, 2, and 3 containing US-treated AP and CSS with their respective formulations containing non-US treated AP and CSS. US = ultrasound treatment; AP = apple pomace; and CSS = coffee silver skin. a–e: Mean values with different superscripts within a chart are statistically different (*p* < 0.05).

**Figure 2 foods-11-02763-f002:**
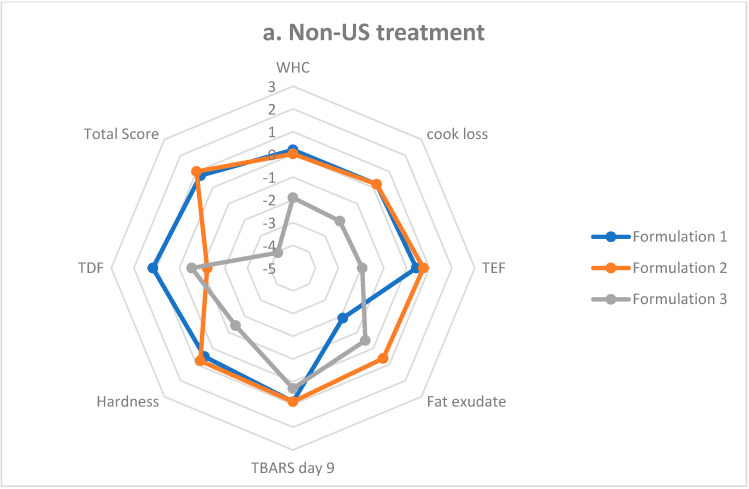
Scores for each of the main physicochemical properties analyzed were reported for sausage Formulations 1–3. (**a**) Non-US treated AP and CSS in sausage formulations; (**b**) US-treated AP and CSS in sausage formulations. The total score for each treatment (divided by 2) was presented. US = ultrasound treatment; AP = apple pomace; and CSS = coffee silver skin.

**Table 1 foods-11-02763-t001:** Mean and standard deviation values of proximate compositions, emulsion stability, and NMR population distribution percentage of sausage formulations. (n = 27).

	Formulation 1			Formulation 2			Formulation 3			Interaction Sig. *
	Non-US Treated AP and CSS	US-Treated AP and CSS	Sig. *	Non-US Treated AP and CSS	US-Treated AP and CSS	Sig. *	Non-US Treated AP and CSS	US-Treated AP and CSS	Sig. *	
**Moisture** (%)	62.8 ± 0.5	62.3 ± 0.9	ns	62.6 ± 0.5	62.7 ± 0.5	ns	63.7 ± 0.6	62.6 ± 0.9	ns	ns
**Fat** (%)	13.3 ± 0.1	13.2 ± 0.3	ns	13.5 ± 0.2	13.5 ± 0.4	ns	12.8 ± 0.5	13.5 ± 0.6	ns	ns
**Protein** (%)	16.0 ± 0.4	16.3 ± 0.7	ns	16.4 ± 0.1	16.0 ± 0.5	ns	16.0 ± 0.5	16.2 ± 0.5	ns	ns
**Ash** (%)	1.8 ± 0.1	1.8 ± 0.1	ns	1.8 ± 0.0	1.8 ± 0.0	ns	1.6 ± 0.0	1.6 ± 0.1	ns	ns
**TDF** (%)	7.6 ± 2.7	6.5 ± 4.0	ns	6.2 ± 2.0	7.8 ± 1.9	ns	6.6 ± 2.7	6.8 ± 1.5	ns	ns
**Salt** (**NaCl**) (%)	0.7 ± 0.1	0.7 ± 0.2	ns	0.9 ± 0.2	0.8 ± 0.2	ns	0.9 ± 0.2	0.8 ± 0.2	ns	ns
**TEF** (%)	8.3 ± 0.3 ^c^	7.7 ± 0.1 ^c^	*	7.5 ± 0.4 ^c^	7.4 ± 0.2 ^c^	ns	13.5 ± 0.6 ^a^	10.6 ± 0.4 ^b^	*	*
**Fat Exudate** (%)	7.4 ± 0.6	5.5 ± 0.2	*	4.3 ± 1.1	4.3 ± 2.9	ns	5.9 ± 1.5	4.4 ± 0.9	ns	ns
**T_2b_** (**ms**)	3.9 ± 0.6	2.5 ± 0.3	*	3.8 ± 0.4	2.4 ± 0.2	*	3.0 ± 0.5	2.1 ± 0.1	*	ns
**T_21_** (**ms**)	36.5 ± 2.1	38.9 ± 0.0	ns	37.7 ± 2.1	37.7 ± 2.1	ns	37.7 ± 2.1	38.9 ± 0.0	ns	ns
**T_22_** (**ms**)	227.3 ± 24.4	257.6 ± 14.1	ns	220.0 ± 21.1	257.6 ± 14.1	ns	234.0 ± 12.8	283.6 ± 15.5	*	ns
**P_2b_** (%)	3.4 ± 0.1 ^a^	2.4 ± 0.1 ^b^	*	3.2 ± 0.4 ^a^	2.4 ± 0.1 ^b^	*	1.8 ± 0.1 ^b^	2.2 ± 0.3 ^b^	ns	*
**P_21_** (%)	89.9 ± 0.6 ^d^	92.1 ± 1.0 ^b,c,d^	*	90.3 ± 0.8 ^c,d^	92.3 ± 0.1 ^a,b,c^	*	94.6 ± 1.2 ^a^	93.2 ± 1.2 ^a,b^	ns	*
**P_22_** (%)	6.8 ± 0.6 ^a^	5.5 ± 1.0 ^a,b,c^	ns	6.5 ± 0.8 ^a,b^	5.3 ± 0.1 ^a,b,c^	ns	3.5 ± 1.0 ^b,c^	4.5 ± 0.9 ^c^	ns	*

US = ultrasound treatment; AP = apple pomace; CSS = coffee silver skin; TDF = total dietary fiber; TEF = total expressible fluid; T_2x_ = peak time distribution values of respective peaks; and P_2x_ = population distribution under the respective curves. * Significance level at *p* < 0.05, ns = not significant. ^a–d^: Mean values with different superscripts within a row are statistically different (*p* < 0.05).

**Table 2 foods-11-02763-t002:** Mean and standard deviation values of color, textural parameters, and TBARS analysis of sausage formulations; n = 27 for color, TBARS; n = 45 for texture.

		Formulation 1			Formulation 2			Formulation 3			Interaction Sig. *
		Non-US Treated AP and CSS	US-Treated AP and CSS	Sig. *	Non-US Treated AP and CSS	US-Treated AP and CSS	Sig. *	Non-US Treated AP and CSS	US-Treated AP and CSS	Sig. *	
**L***		61.8 ± 2.0	61.3 ± 1.8	ns	59.9 ± 1.5	59.9 ± 2.1	ns	64.9 ± 2.2	64.1 ± 1.0	ns	ns
**a***		6.3 ± 0.2	6.6 ± 0.4	ns	6.5 ± 0.1	6.9 ± 0.2	*	6.3 ± 0.4	6.5 ± 0.2	ns	ns
**b***		18.4 ± 1.3	18.9 ± 1.3	ns	18.9 ± 0.7	19.4 ± 0.8	ns	20.1 ± 1.3	20.1 ± 0.6	ns	ns
**ΔE**			0.82			0.60			0.89		
**Hardness** (N)		26.6 ± 5.0	27.7 ± 2.6	ns	27.4 ± 8.4	27.2 ± 3.7	ns	20.7 ± 1.7	20.8 ± 3.2	ns	ns
**Chewiness** (J)		40.2 ± 21.8	31.4 ± 6.9	ns	38.0 ± 21.6	33.7 ± 14.7	ns	14.2 ± 1.0	13.0 ± 2.8	ns	ns
**Cohesive force** (no unit)		0.7 ± 0.0	0.7 ± 0.0	ns	0.8 ± 0.0	0.8 ± 0.0	ns	0.8 ± 0.0	0.8 ± 0.0	ns	ns
**Springiness** (mm)		6.6 ± 0.3	6.4 ± 0.3	ns	6.7 ± 0.9	6.1 ± 0.8	ns	4.5 ± 0.1	4.4 ± 0.3	ns	ns
**TBARS** (mg MDA/kg)	**day 0**	0.2 ± 0.0	0.2 ± 0.0	ns	0.2 ± 0.0	0.2 ± 0.0	ns	0.2 ± 0.0	0.2 ± 0.0	ns	ns
	**day 3**	0.2 ± 0.0	0.2 ± 0.0	ns	0.2 ± 0.0	0.3 ± 0.1	ns	0.2 ± 0.0	0.3 ± 0.0	ns	ns
	**day 6**	0.3 ± 0.1	0.3 ± 0.1	ns	0.3 ± 0.1	0.3 ± 0.1	ns	0.4 ± 0.2	0.4 ± 0.1	ns	ns
	**day 9**	0.3 ± 0.0 ^b,c^	0.4 ± 0.0 ^b^	ns	0.3 ± 0.0 ^c^	0.4 ± 0.0 ^b^	*	0.4 ± 0.1 ^b^	0.8 ± 0.1 ^a^	*	*

US = ultrasound treatment; AP = apple pomace; CSS = coffee silver skin; and **ΔE** = color difference. * Significance level at *p* < 0.05, ns = not significant. ^a–c^: Mean values with different superscripts within a row are statistically different (*p* < 0.05)

## Data Availability

Data is contained within the article.

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
