# Peer review of "A Comparative Study on the Effect of Ultrasound-Treated Apple Pomace and Coffee Silverskin Powders as Phosphate Replacers in Irish Breakfast Sausage Formulations"

_foods, 2022, doi:10.3390/foods11182763_

Round 1
Reviewer 1 Report
Dear Authors,
The study fit in the current trend of reformulating foods to obtain healthier products for consumers. In general, the design of the experiment is fair, but some concerns raised form the information indicated in the paper:
1) Sensory analysis is missing. The inclusion of new ingredients can affect the attributes of reformulated foods
2) Microbial growth data is a necessary aspect to determine shelf life
Specific comments
Abstract
Lines 9-12: The information is repeated. Please revised.
Key-words
Please replace “functional properties” by “technological properties”
Introduction
Line 47: to processed meat products such as…
Line 47: Please remove the semicolon after “products as”
Lines 48-50: In order to avoid repetition, please insert parenthesis to cite the examples of functionalities and delete “such as”
Material and methods
Section 2.2. Sausage production: It is necessary to include a statement indicating that control treatments were produced. Please specify the how control formulations differed from treatments
Lines 258-260: The explanation for the selection of criteria with the information provided in the Figure 2 is not clear. Current information is not detailed enough for other researchers replicate this system in future studies. Please disclose the including and excluding criteria, the statistical validation of this method, the interpretation of results (comparison of indexes within a variable), and practical aspects of this system.
Line 267 and 269: Please check the number of equations
Results and Discussion
Lines 326-329: Since US treatment is a primary factor considered in the study, an explanation for the effect of US in TEF (with specific mention to formulations 1 and 3) results is necessary
Lines 335-336: This statement is not clear. There are significant differences among treatments for fat exudate analysis (formulation 1)
Lines 336-339: One way ANOVA indicates with mean values are significant different or not. This statement about the trend for formulation 3 is not correct. Please revise this statement
Lines 473-475. Please revise this statement for clarity. State that significant difference was observed only in treatment 2
Lines 484-485: Please check the signals for delta E
Line 488: “opposites” is not most appropriate term. Please revise
Line 488-491: Is this statistical result presented in Table 2?
Lines 510-514: Please delete these lines from the manuscript
Lines 523-527: Statistical analysis does not support this claim. There are no statistical analysis for formulations
Lines 552-555: Statistical analysis presented in Table 2 does not support this claim
Line 573: “tenderness” was not evaluated. Please revise it
Conclusion
This section seems more like an abstract than a conclusion. There is some discussion and comparison that should be avoided. Please, clearly state the most adequate formulation(s).
Author Response
The authors would like to convey their thanks to the reviewers for their valuable comments. The authors have addressed each comment and felt that the reviewer’s feedback has improved the manuscript for better readability.
The authors also want to mention that the ‘typesetting’ function used by the journal had made unwanted format changes and the inclusion of unwanted information. For example, Figure 100 – L224 was not in the original manuscript that the authors had uploaded. We made our sincere efforts to eliminate those mistakes in the revised manuscript.
Reviewer #1
The study fits in the current trend of reformulating foods to obtain healthier products for consumers. In general, the design of the experiment is fair, but some concerns were raised from the information indicated in the paper:
1) Sensory analysis is missing. The inclusion of new ingredients can affect the attributes of reformulated foods
The absence of sensory analysis of meat products with US-treated ingredients in this paper could be explained by this uncertainty over the safety of consumption of US-treated food ingredients and the lack of official regulations. The application of the US is still limited to laboratory scale in terms of meat processing and the authors have decided not to risk the consumers because of the challenges like developing standardised methodology and control parameters. However, the authors believe that measuring the colour difference, textural and rancidity (lipid oxidation) addressed the above issue to a certain level.
2) Microbial growth data is a necessary aspect to determine shelf life
The main objective of the study is to examine the ability of US-treated ingredients to improve the reduction in the physicochemical properties of sausages that are majorly influenced by phosphates such as WHC, emulsion stability, textural and antioxidant activities. Hence, this study did not concentrate much on the shelf life analysis of the phosphate-reduced sausage formulations. The authors further state that the shelf-life analysis of the sausage products will be studied as a separate paper in the future with the finalized sausage formulations, and further optimized novel processing technological applications.
Specific comments
Abstract
Lines 9-12: The information is repeated. Please revised.
The repeated information has been revised now to give a clear meaning – Line 11 -12
Key-words
Please replace “functional properties” with “technological properties”
The term “functional properties” have now been replaced with “technological properties” in line - 27
Introduction
Line 47: to processed meat products such as…
The authors reject the reviewer’s suggestion to include the term “such” in the sentence because the authors are meaning to describe that the phosphates are added to meat products as emulsifiers, and so on. The authors feel that the inclusion of the term “such” would change the meaning of the sentence.
Line 47: Please remove the semicolon after “products as”
The semicolon after the word “products as” has been removed from the manuscript in line 47
Lines 48-50: To avoid repetition, please insert parenthesis to cite the examples of functionalities and delete “such as”
The authors have accepted the reviewer’s suggestion and hence have deleted the term “such as” and included parenthesis in the sentence to cite the examples of functionalities. Line 48 -50
Material and methods
Section 2.2. Sausage production: It is necessary to include a statement indicating that control treatments were produced. Please specify how to control formulations differed from treatments
The sentence explaining the preparation of control formulations and the condition on which it is distinguished from the treatment formulations have now been included in the methods and materials section – Lines 162-165 and lines 169-173
Lines 258-260: The explanation for the selection of criteria with the information provided in Figure 2 is not clear. Current information is not detailed enough for other researchers to replicate this system in future studies. Please disclose the including and excluding criteria, the statistical validation of this method, the interpretation of results (comparison of indexes within a variable), and practical aspects of this system.
The authors have used this scoring method as a basic approach to compare the standardised values of important physicochemical properties. Hence the model did not involve any statistical validation into account. The authors have now provided the additional information required for researchers to replicate this in future studies. Lines 305-323
Line 267 and 269: Please check the number of equations
The number of equations has been corrected in the manuscript now after the changes suggested by Reviewer #2. –Lines 315-316
Results and Discussion
Lines 326-329: Since US treatment is a primary factor considered in the study, an explanation for the effect of US in TEF (with specific mention to formulations 1 and 3) results is necessary
The authors have now provided the necessary explanation for the reduction in TEF (%) from the perspective of US-treatment effects on AP and CSS. Lines 377-380
Lines 335-336: This statement is not clear. There are significant differences among treatments for fat exudate analysis (formulation 1)
The authors have deleted the statement as it was confusing for the readers and did not affect the discussion part in any way. Line 387-388
Lines 336-339: One-way ANOVA indicates with mean values are significantly different or not. This statement about the trend for formulation 3 is not correct. Please revise this statement
The authors have revised the sentence to give more clarity and the discussion part including the ‘trend analysis using one-way’ has been removed from the manuscript. Lines 388-391
Lines 473-475. Please revise this statement for clarity. State that significant difference was observed only in treatment 2
The sentence has now been revised for more clarity- “The introduction of US-treated AP & CSS produced a significant increase (P<0.05) in a* values only for formulation 2”. Line 574-577
Lines 484-485: Please check the signals for delta E
The authors apologize for the mistake and have now corrected the signals for delta E – Line as (0 < ∆E < 2) for no difference observed for inexperienced observers. Lines 587-588
Line 488: “opposites” is not the most appropriate term. Please revise
The authors understand that there is the sentence is ambiguous the term ‘opposites’ has been replaced from the manuscript with respective formulations with non-US treated AP & CSS. Line 591-592
Line 488-491: Is this statistical result presented in Table 2?
The authors have realized the mistake in the sentence and hence have deleted the marked sentence from the manuscript. The follow-up sentences were changed according to this change. 592-597
Lines 510-514: Please delete these lines from the manuscript
The suggested lines have now been deleted from the manuscript. Line 613-617
Lines 523-527: Statistical analysis does not support this claim. There are no statistical analyses for formulations
The authors have deleted the sentence since the statistical analysis hasn’t been included to support the claim. Lines – 626-632
Lines 552-555: Statistical analysis presented in Table 2 does not support this claim
The authors realized that there was an ambiguity in the sentence. The error has now been corrected to support the statistical analysis presented in Table 2. Line 656-658
Line 573: “tenderness” was not evaluated. Please revise it
The term “tenderness” has been changed to “hardness” in the manuscript in line - 677
Conclusion
This section seems more like an abstract than a conclusion. There is some discussion and comparison that should be avoided. Please, clearly state the most adequate formulation(s).
The authors have reconsidered the reviewer’s suggestion and made sincere efforts to make the conclusions part more interesting and with more insights for future studies. Line 696.

Reviewer 2 Report
A REVIEW REPORT- FOR AUTHORS
Manuscript Title: A comparative study on the effect of ultrasound-treated apple pomace and coffee silverskin powders as phosphate replacers in Irish breakfast sausage formulations
Manuscript ID: Foods-1877780
OVERVIEW
The manuscript under consideration tests the efficiency of ultrasound-treated apple pomace and coffee silverskin powders as phosphate substitutes in Irish breakfast sausage formulations. This article fits well with the scope and aims of the journal and is considered relevant to the field. The manuscript comes as a continuation of a previously published work by the authors that deemed a solid background to the current research. In fact, the manuscript is original and very interesting, hence got my attention, well written with a clear and smooth structure. All sections of the manuscript are perfectly set, but still, some concerns exist; that will be raised to the authors.
MAJOR COMMENTS
§ Most methods performed in this study were not presented in enough detail to facilitate reproducing these tests, rather only references were cited.
§ The mathematical formulae used in this manuscript are not presented in the right way using the proper function of equations.
§ Tables representing the results should be stand-alone and self-explanatory, which is not the case in the manuscript, different terms and symbols need clarification.
§ Some of the cited literature is too old, hence updating is inevitable.
MINOR COMMENTS
Title
The title is clear, indicative, and describes well the content of the manuscript.
Abstract
It’s quite good, summarizes the manuscript, and gives enough information about the content of the paper including the objective, design, and treatments besides the most significant results and a conclusion.
§ P1 - L18: …using 2 way ANOVA…(2 way without “s” as always known and written).
Keywords
Acceptable.
Introduction
The context of the research work is well set in this section, moreover, the importance of the study to the field is highlighted considering the most recent and pertinent literature. The objective of the study is truly justified in light of a previous study by the authors, in addition to other reviewed literature. Generally, this section is clear and well organized.
Methodology
The manuscript is scientifically sound, and the selected design of the experiment is reliable to test the hypothesis. Proper references of previously published methodology are well documented, but still, some need more details for correct and ease of reproducibility.
§ P4 - L167: Why 20 sausages? On what base did you decide your sample size?
§ P4 - L181: What do you mean by “Day 1”? If means the next day after preparation day, so please, make it clear. Moreover, is it possible to run all these tests (in triplicates) with their replications (20) in one day? Please, explain how you could attain this.
§ P4 - L186: …three independent trials…Do you mean 60 sausages for each formulation in three days? At the end, could you explain how calculations of the results were performed? Are the obtained and presented data come as a result of averaging the outcomes of the three days together, or independently?
§ P4 - L202: “All experiments were carried out in triplicates”..This sentence is not clear, is it for each individual formulation (with 20 samples); any test was performed 3 times? Please, clarify.
§ P5 - L204: WHC and Cook loss: Please, for the benefit of other researchers who will reproduce these tests, write a few lines describing the original methods and later modifications.
§ P5 - L208: LF-NMR: Analysis of bound water: The method of analysis isn’t well explained, more details and steps are needed. The relaxation time distribution compartment should be stated here for ease of understanding of the method and presented results later.
§ P5 - L224: The TEF formula is not clear, but rather confusing. I suggest using the function of equation insertion, instead of writing it manually. No available formula for fat exudate %. Moreover, the numbers: (1), (2), (3), and Figure 100, are ambiguous, no reference for them inside the text, please reconsider.
§ P5 - L229: Texture Profile Analysis: Please, give full details of the method; sample preparation, raw or cooked? How? Cores preparation, the texture analyzer used. The test was run on Day 2, why?
§ P5 - L233: “five sausage cores per formulation”…..OR “five sausage cores per replicate per formulation” that ends in (100) replicates?? Please, this point needs an explanation.
§ P5 - L249: Lipid oxidation: Brief details of the method are required.
§ P6 - L255: Please, give a few details about the scoring method.
§ P6 - L262: Please, use the proper function of equations.
Results and Discussions
The results represent all the reported tests described in M&M section. Different reasonable forms of presentation are used to display collected data. They are logically ordered in a simple and clear pattern. On the other side, the obtained results are well summarized and interpreted considering other results from a variety of studies. Bravely, the authors acknowledge the failure of their hypothesis in achieving some of the expected results. Otherwise, minor observations in this section will be pointed out regarding the tables’ contents and results’ discussion.
§ P8 – L307: Please, add the sample size.
§ P8 – L307: Please, either define the acronyms/symbols in the table notes to add clarity for the reader or write the full words. (TDF, TEF, Ts, Ps)
§ P8 – L308: For easiness and clarity of results, I suggest defining only US, AP, and CSS without mentioning the whole formulation’s ingredients and their percentages.
§ P8 – L309: Please, mention only the level of significance without explanation, it’s fully understood from the table arrangement.
§ P8 – L310: The two asterisks “**” are very confusing with the level of significance (p < 0.01) unless you mean it, isn’t it? It will be fair enough to title the column with “Interaction” and use only one asterisk “*” where applicable.
§ P8 – L311: Please, delete the rest of the sentence starting from “from each…….comparison”.
§ P9 – L322: 2 way ANOVA…(2 way without “s” as always known and written). Please, check all that apply.
§ P9 – L328: Only formulation 3 showed significant differences regarding TEF% in response to US-Treated AP and CSS, while both treatments of formulation 1 shared the common superscript letter “C”, Please, reconsider and make the necessary changes accordingly.
§ P9 – L360: The same trend was…..Please, reconsider, as the trend here is not similar. In US-treated AP and CSS, the highest value is reported by F2 followed by F1 and lastly F3, while in non-US treated AP and CSS, the highest value is reported by F1 followed by F2 and lastly F3.
§ P10 – L371: “This increase in WHC…Do you mean an increase in WHC percentage or the ability of sausages to bind water? There is a big difference between the two phrases. Based on your method of WHC determination and calculation, the increase or decrease in WHC has completely different meanings. The increase in WHC percentage indicates a high loss of water ending in poor WHC, and vice versa. Please, reconsider this point and correct it if it deserves.
§ P10 – L376: “in increased water absorption…”, if the percentage value of WHC increased, this indicates the lower capacity to hold water, which is opposite to the statement mentioned. Please, recheck the method and calculations for determining WHC and made changes if necessary.
§ P10 – L378: Please, delete a). and rewrite: 1a. inside the chart area (P10) along with the caption = (1a. WHC of sausage formulations) for clarity and ease of presentation and understanding.
§ P10 – L381: Please, delete b). and rewrite: 1b. inside the chart area (P11) along with the caption = (1b. Cook loss of sausage formulations) for clarity and ease of presentation and understanding.
§ P11 – L386: For simplicity, easiness, and understanding of the chart, I suggest defining only US, AP, and CSS without mentioning the whole formulation’s ingredients and their percentages.
§ P11 – L388: Please, delete the rest of the sentence starting from “from each…….comparison”.
§ P11 – L396: “for all sausage formulations”….This is not true, as F1 showed no significant difference between the two treatments. Please, correct.
§ P12 – L411: Water mobility measured using LF-NMR: The data of this parameter is displayed in Table1, so I was expecting it to be discussed after emulsion stability (3.2), but this was not the situation. It will be more convenient to reorder this section from (3.4) to (3.3) for a better presentation of the results. Then, reordering other sections accordingly.
§ P12 – L452-453: Please, revise these sentences when reordering sections to avoid interpreting results before they are presented.
§ P13 – L482: The values were reported irrespective of their formulations (F1, F2 or F3), please, fix this situation.
§ P13 – L484: I guess the range of the total color difference (∆E) is incorrect, it supposes to be as follows: (0 < ∆E < 1), please, recheck.
§ P14 – L496: Table 2: Please, follow the same recommendations for table 1.
§ P17 – L586: Please, delete (i) and (ii) and rewrite: i inside the chart area (P17) along with the caption = (i. Non-US treatment) for clarity and ease of presentation and understanding.
§ P18 – Chart: Please, rewrite: ii inside the chart area (P18) along with the caption = (ii. US- treatment) for clarity and ease of presentation and understanding.
Conclusions
This part successfully considered the main points of a typical manuscript conclusion. It restates the research objective, pointing out the key results and their importance to the current practical applications. It also addresses the major limitations of the study and concludes this section with a solid statement.
§ P19 – L592: Conclusions: Based on your research outcomes and drawn conclusions, could you recommend increasing the percentage of AP and CSS inclusion beyond the level being used in the tested formulations? Whatever the answer is, could you add it to the conclusions as a roadmap for further studies?
References
The cited literature in this manuscript is up to date (46% of the reviews fall within the past 5 years). They are clear, comprehensive, and relevant to the topic of the study. Nevertheless, some reviews are too old (1920, 1978). Please, update them unless they are basic references and no other recent alternatives.

Author Response
The authors would like to convey their thanks to the reviewers for their valuable comments. The authors have addressed each comment and felt that the reviewer feedback has improved the manuscript for better readability.
The authors also want to mention that the ‘typesetting’ function used by the journal had made unwanted format changes and inclusion of unwanted information. For example, Figure 100 – L224 was not in the original manuscript that the authors had uploaded. We made our sincere efforts to eliminate those mistakes in the revised manuscript.
Reviewer #2
Manuscript Title: A comparative study on the effect of ultrasound-treated apple pomace and coffee silverskin powders as phosphate replacers in Irish breakfast sausage formulations
Manuscript ID: Foods-1877780
OVERVIEW
The manuscript under consideration tests the efficiency of ultrasound-treated apple pomace and coffee silverskin powders as phosphate substitutes in Irish breakfast sausage formulations. This article fits well with the scope and aims of the journal and is considered relevant to the field. The manuscript comes as a continuation of a previously published work by the authors that deemed a solid background to the current research. The manuscript is original and very interesting, hence got my attention, well written with a clear and smooth structure. All sections of the manuscript are perfectly set, but still, some concerns exist; that will be raised to the authors.
MAJOR COMMENTS
- Most methods performed in this study were not presented in enough detail to facilitate reproducing these tests, rather only references were cited.
As per the reviewer’s suggestion, more detailed methods have now been provided for all the physicochemical analyses used in the manuscript that would allow future researchers to reproduce these tests.
- The mathematical formulae used in this manuscript are not presented in the right way using the proper function of equations.
The authors apologise for not using the insert equation option to describe various equations used in the manuscript. The error has now been corrected by converting all the equations using the ‘insert equation’ option.
- Tables representing the results should be stand-alone and self-explanatory, which is not the case in the manuscript, different terms and symbols need clarification.
The authors acknowledge the review’s claim and hence have removed most of the unwanted information in the table notes. The authors have also clarified the various terms and symbols used in the table and expanded them in the table notes.
- Some of the cited literature is too old, hence updating is inevitable.
The authors have reconsidered the reviewer’s comments and made sincere efforts to replace the old reference with the new one. However, the authors have untouched the reference in the methods and materials since they are the original references. Along with that, the authors have decided to keep the reference Figuerola (2005), since the information given is more original to this paper. The references were re-ordered and re-numbered according to the changes and all the track changes pertaining to references were accepted for our convenience.
MINOR COMMENTS
Title
The title is clear, indicative, and describes well the content of the manuscript.
Abstract
It’s quite good, summarizes the manuscript, and gives enough information about the content of the paper including the objective, design, and treatments besides the most significant results and a conclusion.
- P1 - L18: …using 2-way ANOVA (2 way without “s” as always known and written).
The above-mentioned mistake has been corrected in the whole manuscript by changing the word into ‘2-way ANOVA’. Line 19
Keywords
Acceptable.
Introduction
The context of the research work is well set in this section, moreover, the importance of the study to the field is highlighted considering the most recent and pertinent literature. The objective of the study is truly justified in light of a previous study by the authors, in addition to other reviewed literature. Generally, this section is clear and well organized.
Methodology
The manuscript is scientifically sound, and the selected design of the experiment is reliable to test the hypothesis. Proper references of previously published methodology are well documented, but still, some need more details for correct and ease of reproducibility
*The following three comments will be answered together*
- P4 - L167: Why 20 sausages? On what base did you decide your sample size?
- P4 - L186: …three independent trials…Do you mean 60 sausages for each formulation in three days? In the end, could you explain how calculations of the results were performed? Are the obtained and presented data come as a result of averaging the outcomes of the three days together, or independently?
- P4 - L181: What do you mean by “Day 1”? If means the next day after preparation day, so please, make it clear. Moreover, is it possible to run all these tests (in triplicates) with their replications (20) in one day? Please, explain how you could attain this.
The authors have removed the number 20 from the manuscript to avoid confusion. The experimentation model is explained as follows: Approximately 30 (20 was a mistake) sausages per formulation were prepared per formulation and they are randomly assigned for various experiments per se WHC & Cook loss (3 sausages); texture (5 sausages); emulsion stability (3 sausages); TBARS (12 – 3 sausages per day 0, 3, 6 & 9); Colour & NMR (3 sausages).
This means that for a single independent trial, approximately 30 sausages per formulation were produced. For WHC, three sausages were randomly chosen and packed on day 0. On day 1 (The next day of preparation day), three sausages were analysed for WHC values in triplicates.
The obtained presented data in this study was based on the result of averaging the outcomes of three days together.
These entire issues have been addressed in the manuscript now – Sample size per analysis – Line 185 -191
Average of three independent trials – Line 194-195; day 1 – Line 185
- P4 - L202: “All experiments were carried out in triplicates.” This sentence is not clear, is it for each individual formulation (with 20 samples); any test was performed 3 times? Please, clarify.
The authors use the above-explained answer for this comment and state that the experiments were carried out in three repetitions per independent trial and the values were averaged and displayed. This has now been explained in the manuscript in lines 210-211
- P5 - L204: WHC and Cook loss: Please, for the benefit of other researchers who will reproduce these tests, write a few lines describing the original methods and later modifications.
A more detailed scientific procedure to calculate WHC and cook loss has now been provided in the manuscript. Lines – 214-230
- P5 - L208:LF-NMR: Analysis of bound water: The method of analysis isn’t well explained, more details and steps are needed. The relaxation time distribution compartment should be stated here for ease of understanding of the method and presented results later.
A more detailed method of analysis has been added to the manuscript and the introduction to relaxation time distribution has been provided now. Lines 235-244
- P5 - L224: The TEF formula is not clear, but rather confusing. I suggest using the function of equation insertion, instead of writing it manually. No available formula for fat exudate %. Moreover, the numbers: (1), (2), (3), and Figure 100, are ambiguous, no reference for them inside the text, please reconsider.
The author apologies for the mode of the equation and hence the formulas are now developed using the ‘insert equation’ option. The formula for fat exudate % has been removed from the manuscript during the ‘typesetting’ formatting used by the journal. The same applies to the term Figure 100. This was not in the original manuscript submitted. The entire ambiguity has now been removed from the manuscript and a clear equation number was provided. Lines 253-257
*The following two comments will be answered together*
- P5 - L229: Texture Profile Analysis: Please, give full details of the method; sample preparation, raw or cooked? How? Cores preparation, the texture analyzer used. The test was run on Day 2, why?
- P5 - L233: “five sausage cores per formulation”…..OR “five sausage cores per replicate per formulation” that ends in (100) replicates?? Please, this point needs an explanation.
The full details of the texture profile analysis procedure have been given in the manuscript and the reason for performing the analysis on day 2 was also provided (day 1 cooking; day 2 – analysis) in lines – 259-272
The explanation for the repetitions has now been provided in lines – 259, 262-263
- P5 - L249: Lipid oxidation: Brief details of the method are required.
Details of TBARS analysis have been provided in lines 292-301.
- P6 - L255: Please, give a few details about the scoring method.
Additional information about the scoring methods has now been provided in the manuscript. Lines 305-323
- P6 - L262: Please, use the proper function of equations.
Proper equation functions have been used now to provide the scoring formula. Line 310-311
Results and Discussions
The results represent all the reported tests described in M&M section. Different reasonable forms of presentation are used to display collected data. They are logically ordered in a simple and clear pattern. On the other side, the obtained results are well summarized and interpreted considering other results from a variety of studies. Bravely, the authors acknowledge the failure of their hypothesis in achieving some of the expected results. Otherwise, minor observations in this section will be pointed out regarding the tables’ contents and results’ discussion.
*The following six comments will be answered together*
- P8 – L307: Please, add the sample size.
- P8 – L307: Please, either define the acronyms/symbols in the table notes to add clarity for the reader or write the full words. (TDF, TEF, Ts, Ps)
- P8 – L308: For easiness and clarity of results, I suggest defining only US, AP, and CSS without mentioning the whole formulation’s ingredients and their percentages.
- P8 – L309: Please, mention only the level of significance without explanation, it’s fully understood from the table arrangement.
- P8 – L310: The two asterisks “**” are very confusing with the level of significance (p < 0.01) unless you mean it, isn’t it? It will be fair enough to title the column with “Interaction” and use only one asterisk “*” where applicable.
- P8 – L311: Please, delete the rest of the sentence starting from “from each…….comparison”.
All the above-mentioned comments have been acknowledged by the authors and changed as per the suggestion. Sample size and abbreviations have been provided and the unwanted information was removed. Lines – 358-363
- P9 – L322: 2-way ANOVA… (2 way without “s” as always known and written). Please, check all that apply.
The term “2-ways ANOVA” has been changed to “2-way ANOVA” in the whole manuscript.
- P9 – L328: Only formulation 3 showed significant differences regarding TEF% in response to US-Treated AP and CSS, while both treatments of formulation 1 shared the common superscript letter “C”, Please, reconsider and make the necessary changes accordingly.
The authors understand that there is an ambiguity in the suggested sentence and have now corrected the sentence by showing that the results were based on the one-way ANOVA analysis between the individual formulations. A more detailed explanation was now provided in the lines 375
- P9 – L360: The same trend was…..Please, reconsider, as the trend here is not similar. In US-treated AP and CSS, the highest value is reported by F2 followed by F1 and lastly F3, while in non-US treated AP and CSS, the highest value is reported by F1 followed by F2 and lastly F3.
The authors have accepted the reviewer’s suggestion and hence have changed the manuscript accordingly to give a clear meaning. Line 455-461
*The following two comments will be answered together*
- P10 – L371: “This increase in WHC…Do you mean an increase in WHC percentage or the ability of sausages to bind water? There is a big difference between the two phrases. Based on your method of WHC determination and calculation, the increase or decrease in WHC has completely different meanings. The increase in WHC percentage indicates a high loss of water ending in poor WHC, and vice versa. Please, reconsider this point and correct it if it deserves.
- P10 – L376: “in increased water absorption…”, if the percentage value of WHC increased, this indicates the lower capacity to hold water, which is opposite to the statement mentioned. Please, recheck the method and calculations for determining WHC and made changes if necessary.
The authors reject the reviewer’s hypothesis using the method of WHC determination used in this study.
The formula is as follows
B- the initial weight of the sample; A- centrifuged weight; M- total moisture content; In here, ((B-A)/M) gives the total water lost as a result of the cooking and centrifugation process. So, (B-A)/M x 100 would be the percentage of water lost from the sausages. However, the formula involves 1- ((B-A/M) x100, which gives the percentage of water not lost, i.e. percentage of water retained in the sausages. Hence, the higher the percentage values, the higher the water retained in the sausages. Thus, an increase in WHC values means an increase in the ability of sausages to hold more water.
In the second comment, the authors meant that the increase in WHC due to the addition of US-treated AP and CSS was because of the increase in water absorption capacity values of US-treated AP and CSS which was proved in our previously published paper. 20. Thangavelu KP, Tiwari BK, Kerry JP, Álvarez C (2022). A comprehensive study on the characterisation properties of power ultrasound-treated apple pomace powder and coffee silverskin powder. European Food Research and Technology 248:1939-1949. This has been clearly explained in the manuscript now.
- P10 – L378: Please, delete a). and rewrite: 1a. inside the chart area (P10) along with the caption = (1a. WHC of sausage formulations) for clarity and ease of presentation and understanding.
The authors have accepted the reviewer’s suggestion and have included “1a” inside the chart area and now it reads as “1a. WHC of sausage formulations” – Line 480
- P10 – L381: Please, delete b). and rewrite: 1b. inside the chart area (P11) along with the caption = (1b. Cook loss of sausage formulations) for clarity and ease of presentation and understanding.
The authors have accepted the reviewer’s suggestion and have included “1b” inside the chart area and now it reads as “1b. Cook loss of sausage formulations” – Line 483
- P11 – L386: For simplicity, easiness, and understanding of the chart, I suggest defining only US, AP, and CSS without mentioning the whole formulation’s ingredients and their percentages.
The authors have accepted the reviewer’s suggestion and have removed the unwanted sentences. Line 488
- P11 – L388: Please, delete the rest of the sentence starting from “from each…….comparison”.
The authors have accepted the reviewer’s suggestion and have removed the unwanted sentences. Lines 491
- P11 – L396: “for all sausage formulations”….This is not true, as F1 showed no significant difference between the two treatments. Please, correct.
The author apologises for the mistake and hence has corrected the mistake by removing formulation 1 from significant terms. Lines 497-499
*The following two comments will be answered together*
- P12 – L411: Water mobility measured using LF-NMR: The data of this parameter is displayed in Table1, so I was expecting it to be discussed after emulsion stability (3.2), but this was not the situation. It will be more convenient to reorder this section from (3.4) to (3.3) for a better presentation of the results. Then, reordering other sections accordingly.
- P12 – L452-453: Please, revise these sentences when reordering sections to avoid interpreting results before they are presented.
As per the reviewer’s suggestion, the section explaining the water mobility measured using LF-NMR was moved above the WHC and cook loss section. The sections were reordered accordingly. The sentences explaining the WHC and cook loss values in LF-NMR have been removed to avoid interpreting the results before they are presented. Lines – 403
The references have also been rearranged and “accept change” –ed in the track changes for our convenience.
- P13 – L482: The values were reported irrespective of their formulations (F1, F2 or F3), please, fix this situation.
The authors acknowledge the ambiguity in the given information and hence have added the formulation numbers before their respective numerical data. Lines – 584-585
- P13 – L484: I guess the range of the total colour difference (∆E) is incorrect, it supposes to be as follows: (0 < ∆E < 1), please, recheck.
The authors apologize for the mistake and have now corrected the signals for delta E as (0 < ∆E < 2) for no difference observed for inexperienced observers. Lines 587-589
- P14 – L496: Table 2: Please, follow the same recommendations for table 1.
The above accepted and rejected changes for Table 1 have been applied to Table 2. Line - 600
- P17 – L586: Please, delete (i) and (ii) and rewrite: i inside the chart area (P17) along with the caption = (i. Non-US treatment) for clarity and ease of presentation and understanding.
The authors have accepted the reviewer’s suggestion and have included “i” inside the chart area and now it reads as “i. Non-US treatment.” Line - 690
- P18 – Chart: Please, rewrite: ii inside the chart area (P18) along with the caption = (ii. US- treatment) for clarity and ease of presentation and understanding.
The authors have accepted the reviewer’s suggestion and have included “ii” inside the chart area and now it reads as “ii. US treatment.” – Line 691
Conclusions
This part successfully considered the main points of a typical manuscript conclusion. It restates the research objective, pointing out the key results and their importance to the current practical applications. It also addresses the major limitations of the study and concludes this section with a solid statement.
- P19 – L592: Conclusions: Based on your research outcomes and drawn conclusions, could you recommend increasing the percentage of AP and CSS inclusion beyond the level being used in the tested formulations? Whatever the answer is, could you add it to the conclusions as a roadmap for further studies?
The authors acknowledge the reviewer’s comments and hence have included the sentences in the conclusions section focusing on future studies that could be derived from this study. Lines 696

Reviewer 3 Report
The paper has interesting data obtained through the use of well-selected established and proved methods and analyses. Moreover, the paper covers very topical issue and provides data that will complement the data from previously published research by these and other authors.
However, the authors should make some minor improvements.
Line 228: what is figure 100? Where is equation (3)?
Line 230: Bourne M (2002) in Food Texture and Viscosity: Concept and Measurement, 2nd ed, on p184 stated: …”Therefore, in reporting TPA measurements one should report either chewiness values, or gumminess values but not both for the same food.”
Lines 267–272: Equations 7–9 are missing.
Lines 274: Why formulation was not observed as a factor?
Line 378: Figure 1 should be repair – small letters should be closer to the corresponding bars.
Lines 492–493: This statement is just guessing since there are no significant differences
Line 523–529: The authors discussed about influence of formulation on TPA properties but did not include formulation in statistical analysis!
Line 528: firmness is not examined!
Author Response
The authors would like to convey their thanks to the reviewers for their valuable comments. The authors have addressed each comment and felt that the reviewer’s feedback has improved the manuscript for better readability.
The authors also want to mention that the ‘typesetting’ function used by the journal had made unwanted format changes and the inclusion of unwanted information. For example, Figure 100 – L224 was not in the original manuscript that the authors had uploaded. We made our sincere efforts to eliminate those mistakes in the revised manuscript.
The paper has interesting data obtained through the use of well-selected established and proven methods and analyses. Moreover, the paper covers very topical issue and provides data that will complement the data from previously published research by these and other authors.
However, the authors should make some minor improvements.
- Line 228: what is figure 100? Where is equation (3)?
As explained above, it was a mistake caused by the typesetting formatting by the journal. The error has been corrected and the proper formula is provided for equation (3). Line 257
- Line 230: Bourne M (2002) in Food Texture and Viscosity: Concept and Measurement, 2nded, on p184 stated: …”Therefore, in reporting TPA measurements one should report either chewiness values, or gumminess values but not both for the same food.”
The authors thank the reviewers for pointing out this mistake in the manuscript. We realised that some authors suggest that chewiness is reported for solid foods and gumminess is for semisolid foods, therefore mutually excluding. Based on that, we have excluded gumminess for this report and discussing only the results related to chewiness.
- Lines 267–272: Equations 7–9 are missing.
The error has been now corrected and the equations numbers are reordered as per the other reviewer’s comments. Line 310
- Lines 274: Why formulation was not observed as a factor?
Line 332- The authors have used ingredients, formulations, and ingredients*formulations as the factor for analysis 2-way ANOVA. In addition, one-way ANOVA was carried out in the individual formulations with US-treatment as a factor. All the results were based on both analyses and we trust this is clearly explained in the manuscript.
- Line 378: Figure 1 should be repaired – small letters should be closer to the corresponding bars.
The authors acknowledge the reviewer’s comment and have made the necessary changes in Figure 1 by moving the superscripts closer. Lines 478-480
- Lines 492–493: This statement is just guessing since there are no significant differences
The authors acknowledge the mistake in the manuscript and hence have removed the sentence from the manuscript based on the reviewer’s and other reviewer’s comments. Lines 592-599
- Line 523–529: The authors discussed about influence of formulation on TPA properties but did not include formulation in statistical analysis!
As explained above, formulations were considered as factor for 2-way ANOVA analysis and the discussions made in the manuscript was based on the 2-way ANOVA analysis results.
- Line 528: firmness is not examined!
The authors have removed the sentence involving the term “firmness” during the correction suggested by the other two reviewers. Hence, the term is not present anymore in the manuscript. Line. 631

Round 2
Reviewer 1 Report
Dear Authors,
The paper was properly revised and now is suitable for publication.